# Organ sculpting by patterned extracellular matrix stiffness

**Justin Crest[1], Alba Diz-Muñoz[2†], Dong-Yuan Chen[1], Daniel A Fletcher[2], David Bilder[1*]**

[1]Department of Molecular and Cell Biology, University of California-Berkeley, Berkeley, United States; [2]Department of Bioengineering and Biophysics Program, University of California-Berkeley, Berkeley, United States

**Abstract** How organ-shaping mechanical imbalances are generated is a central question of morphogenesis, with existing paradigms focusing on asymmetric force generation within cells. We show here that organs can be sculpted instead by patterning anisotropic resistance within their extracellular matrix (ECM). Using direct biophysical measurements of elongating Drosophila egg chambers, we document robust mechanical anisotropy in the ECM-based basement membrane (BM) but not in the underlying epithelium. Atomic force microscopy (AFM) on wild-type BM in vivo reveals an anterior–posterior (A–P) symmetric stiffness gradient, which fails to develop in elongation-defective mutants. Genetic manipulation shows that the BM is instructive for tissue elongation and the determinant is relative rather than absolute stiffness, creating differential resistance to isotropic tissue expansion. The stiffness gradient requires morphogen-like signaling to regulate BM incorporation, as well as planar-polarized organization to homogenize it circumferentially. Our results demonstrate how fine mechanical patterning in the ECM can guide cells to shape an organ.

**\*For correspondence:** bilder@berkeley.edu

**Present address:** [†]Cell Biology and Biophysics Unit, European Molecular Biology Laboratory, Heidelberg, Germany

**Competing interests:** The authors declare that no competing interests exist.

## Introduction

Animal organs have a bewildering variety of distinctive forms that are critical for their functions. Although originating in a genetic program, morphogenesis of organs ultimately depends on physical forces, and specifically on their imbalances, to drive shape change (*Thompson, 1917*). A central question of morphogenesis is how such force imbalances are created by mechanical anisotropy that is generated within an organ's components. Current paradigms derive from archetypes of morphogenetic processes such as tissue elongation, and elegant studies have revealed conserved mechanisms that drive elongation across many species. In the *Drosophila* embryo, planar cell polarized (PCP) myosin contractility at the cell cortex generates junctional rearrangements that extend the germband, whereas in vertebrate embryos, PCP actin-based protrusions drive cell movements that extend the neural plate (*Guillot and Lecuit, 2013*; *Heisenberg and Bellaïche, 2013*; *Vichas and Zallen, 2011*; *Walck-Shannon and Hardin, 2014*). In these textbook examples of morphogenesis, as in others such as gastrulation and epiboly, the force anisotropies that instruct shape are generated within the tissue's cells.

In theory, asymmetric organs could be generated not only by spatially varying forces produced within cells, but also by spatially varying tissue properties that differentially resist uniformly applied forces. In epithelial organs, morphogenetic forces include not only tension between cells that can cause intercellular rearrangements, but also expansion of luminal contents normal to the epithelial plane; resistance to these forces is mediated by cells and by the extracellular matrix (ECM), including the basement membranes (BMs) that line all epithelia. In comparison to the action of cellular forces, the role of non-cellular influences on morphogenesis is poorly understood.

**eLife digest** All organs have specific shapes and architectures that are necessary for them to work properly. Many different factors are responsible for arranging the right cells into the correct positions to make an organ. These include physical forces that act within and around cells to pull them into the right shape and location.

A structure called the extracellular matrix surrounds cells and provides them with support; it can also guide cell movements. It is not clear whether the extracellular matrix plays only a passive role or a more active, instructive role in shaping organs, in part, because it is difficult to measure the physical forces within densely packed cells.

The ovaries of the fruit fly *Drosophila melanogaster* provide a simple system in which to study how organs take their shape. Crest et al. developed a method to measure forces in the fly ovary as it changes from being an initially spherical group of cells to its final elongated tube shape. The results revealed that, during this process, the extracellular matrix becomes gradually stiffer from one end of the ovary to the other. This change is the main factor responsible for the cell rearrangements that shape the developing organ.

This work reveals that, along with providing structural support to cells, the mechanical properties of the matrix also actively guide how organs form. In the future, these findings may aid efforts to grow organs in a laboratory and to regenerate organs in human patients.

A comprehensive study of morphogenetic mechanics requires a tissue that is subject to both cellular and extracellular influences. The *Drosophila* egg chamber (or 'follicle') is such a tissue (*Figure 1A* and *Figure 1—figure supplement 1*) and undergoes robust elongation during its development (*Spradling, 1993*). Each follicle is a simple tube-like organ consisting of just two cell types, with a somatic epithelium of 'follicle cells' (FCs) encasing an interconnected cyst of germ cells. The epithelium also produces an underlying BM that surrounds the entire follicle. The organ is initially spherical and grows throughout oogenesis, expanding ~5000 fold in volume over ~3 days. Expansion for the first 35 hr is isotropic, but subsequently becomes anisotropic as the follicle elongates >2-fold specifically along the anterior–posterior (A–P) axis to form the distinctively shaped oval egg (*Figure 1A*). Much of this elongation takes place without cell division. Genes and cell behaviors that are required for egg elongation have been identified, but the mechanical environment that actually shapes the tissue is not known (*Bilder and Haigo, 2012*; *Cetera and Horne-Badovinac, 2015*).

Here we use biophysical tools to measure the mechanical conditions present in elongating follicles. Surprisingly, we find no evidence for differential cell-intrinsic forces within the organ, but instead document a robust spatial gradient in stiffness within the BM. Direct BM manipulation indicates that this mechanical gradient is instructive for tissue elongation. Fine mechanical patterning within the BM, generated by independent mechanisms along both the A–P and circumferential axes, endows the BM with anisotropic resistance to tissue expansion that deforms the growing tissue. These results highlight a new parameter of developmental mechanics by uncovering an unappreciated sophistication in BM mechanical properties that can directly impose organ shape.

## Results

### Cells in elongating follicles are mechanically isotropic

To understand the conditions that drive elongation of the *Drosophila* follicle, we first searched for mechanical anisotropy in the organ's two distinct cell populations. In these assays as well as others below, we examined follicles at stage 8 and earlier, when they display a regular and A–P symmetric morphology. Previous genetic mosaic experiments with several 'round egg' mutations exclude the germline as a site of action (*Frydman and Spradling, 2001*; *Wieschaus et al., 1981; Viktorinová et al., 2009*), while stripping of epithelium in *Heteropeza* results in round rather than elongated follicles (*Went and Junquera, 1981*). Similarly, we genetically ablated the *Drosophila* follicle epithelium (as well as its underlying BM), and found that germline growth resulted in a nearly spherical follicle at stages when elongation would normally have initiated (*Figure 1—figure*

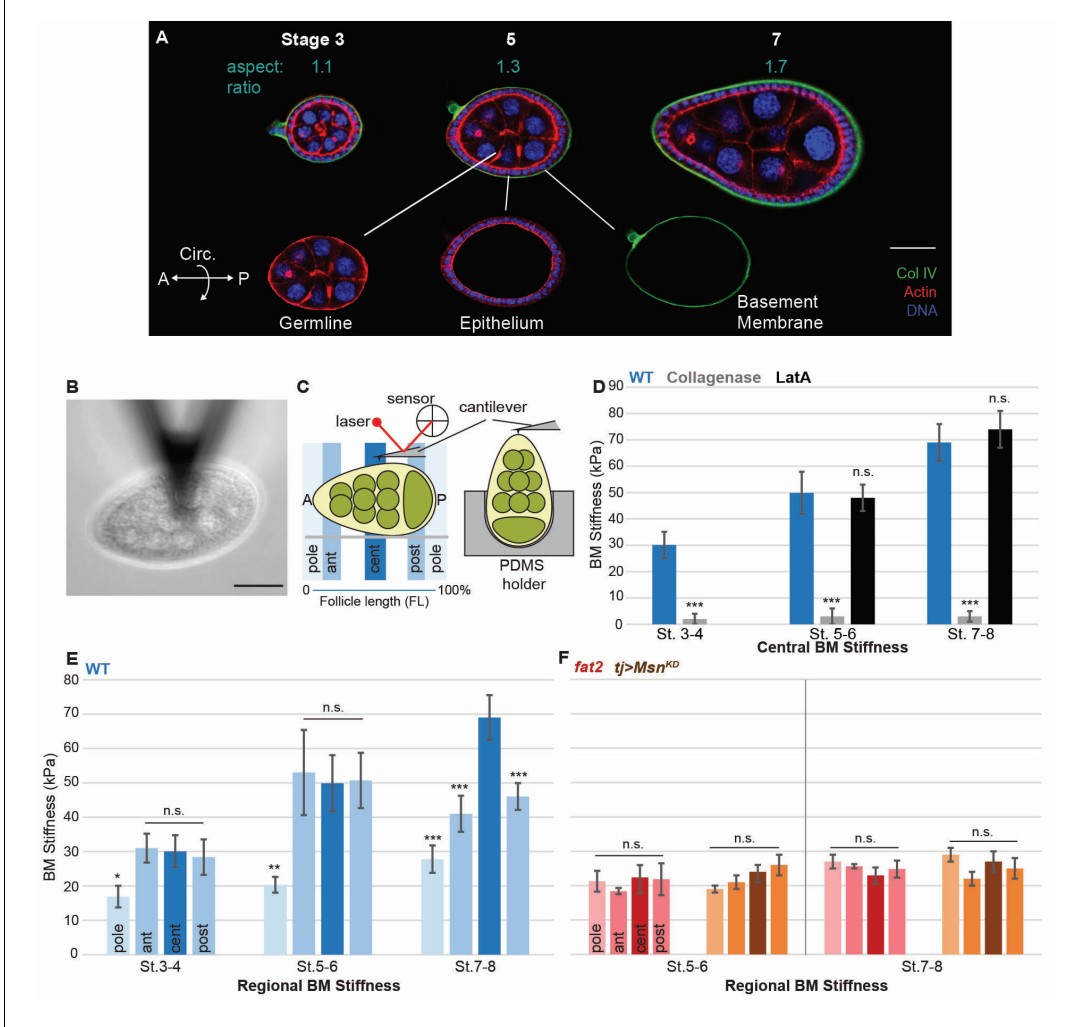

**Figure 1.** A mechanical stiffness gradient in the follicle basement membrane. (A) Elongation of the *Drosophila* follicle during oogenesis involves three components: the luminal germline, a surrounding epithelium, and an encasing basement membrane (BM) (see also *Figure 1—figure supplement 1*). Aspect ratios of stage 3, 5, and 7 egg chambers stained for DAPI (blue) and phalloidin (red), along with ColIV–GFP (green), are shown. (B) Atomic Force Microscopy (AFM) measurement of BM stiffness in living follicles. Absence of stroma and external position of BM allow direct access of the AFM probe. (C) Follicles are probed at different regions along the A–P axis, including the poles via Polydimethylsiloxane (PDMS) 'egg cartons'. Stiffness measurements are derived from the first 50 nm of force–extension curves. (D) BM stiffness in the follicle center increases during development. Collagen digestion but not F-actin network disruption eliminates nearly all AFM-measured stiffness. (cf *Figure 1—figure supplement 1*). (E) Regional BM stiffness along the follicle A–P axis; color intensity matches position as in(C). WT follicles develop an A–P symmetrical gradient of mechanical anisotropy. Anterior and posterior poles are not distinguished. (F) *fat2*- and *msn*-depleted follicle BMs do not increase stiffness during development and remain mechanically isotropic. Scale bar: 25 μm.

The following figure supplements are available for figure 1:

**Figure supplement 1.** Isotropic mechanical properties of cells in the *Drosophila* ovary.

**Figure supplement 2.** AFM elasticity measurement method.

**Figure supplement 3.** Validation of pharmacological and hypertonic shock treatments for BM stiffness.

*supplement 1*). Together, these data suggest that the germline is not an intrinsic source of mechanical anisotropy.

To assess whether the follicle epithelium showed PCP cortical contractility, we laser-ablated cellular junctions at different positions along the A–P axis and measured the recoil. In elongating

epithelia including the *Drosophila* ectoderm and wing, this technique reveals differential tension along A–P and dorsal–ventral (D–V) axes, an anisotropy associated with polarized Myosin II accumu–lation (*Bosveld et al., 2012*; *Etournay et al., 2015*; *Fernandez-Gonzalez et al., 2009*; *Rauzi et al., 2008*). However, in the elongating follicle epithelium, dissection of junctions resulted in equivalent retraction of A–P and circumferentially oriented junctions; polarized accumulation of Myo:GFP was not observed (*Figure 1—figure supplement 1*). These results suggest that neither follicle cell type intrinsically generates anisotropic physical forces.

## Patterned mechanical stiffness in the follicle BM

To identify the source of mechanical anisotropy, we therefore turned to a non-cellular component of the organ: the ECM, specifically the BM. The *Drosophila* follicle is enclosed by a BM that, like classic vertebrate BMs, is ~150 nm thick and contains Collagen IV, laminin, and perlecan (*Haigo and Bilder, 2011*; *Isabella and Horne-Badovinac, 2015*; *Spradling, 1993*). BMs and surrounding ECM are known to have important influences on animal organogenesis (*Daley and Yamada, 2013*; *Morrissey and Sherwood, 2015*), but discovery of their mechanical roles has been impeded by the difficulty of measuring these directly in vivo. In the *Drosophila* follicle, the external position of the BM, the absence of a cellular stroma (*Figure 1A* and *Figure 1—figure supplement 1*) and the ability to develop in culture provided an unprecedented opportunity to assess the mechanical properties of an intact BM, in living tissue under physiological conditions.

We utilized Atomic Force Microscopy (AFM) to measure BM stiffness, calculating the Young's modulus from the deflection of a cantilevered probe indenting into the basal follicle surface (*Figure 1B,C*, *Figure 1—figure supplement 2*). Treatment of follicles with purified collage-nase decreased stiffness by 97% without detectable changes to epithelial junctions, whereas disrup-tion of the cellular actomyosin network with Latrunculin A induced no significant change in the AFM measurements. Furthermore, reducing the turgor pressure of the follicle with a hypertonic solution (2000 mOsm sorbitol media) does not have an effect on the BM stiffness (*Figure 1D* and *Figure 1—figure supplement 3*). These controls indicate that the quantified stiffness predominantly derives from the BM.

AFM measurements at the center of staged wild-type (WT) follicles showed that the BM gradually stiffens as oogenesis proceeds, increasing from ~30 KPa at stage 3 to ~40 KPa at stage 5 and ~70 KPa at stage 7 (*Figure 1D*). Interestingly, although stiffness was highly consistent (>5% variance) around the circumferential axis at a given position Figure 4F, it significantly varied along the A–P axis (*Figure 1E*). At stages 3 and 5, poles were ~50% softer than the central or terminal regions (see *Figure 1D* for definitions). This difference persisted into later stages, and the central regions further became ~30% stiffer than the terminal regions. Thus, AFM analysis reveals a symmetrical gradient of BM stiffness along the A–P axis of the follicle.

## BM stiffness is instructive for tissue elongation

If the BM stiffness gradient is functionally important for organ elongation, it should be perturbed in conditions where elongation fails. We analyzed two distinct genotypes in which follicle elongation is defective: mutants for *fat2*, which encodes an atypical cadherin that controls basal PCP organization in the follicle epithelium (*Viktorinová et al., 2009*), and RNAi-depleting mutants for *misshapen* (*msn*), which encodes a kinase that negatively regulates integrin-mediated adhesion (*Lewellyn et al., 2013*). We carried out AFM on staged *fat2* follicles and found that, unlike WT follicles, BM stiffness did not increase from stage 5 to stage 7 (*Figure 1F*). Strikingly, *fat2* follicles showed no significant differences between the central, terminal, and polar regions at any stage. An isotropic and softer BM was also seen in *msn*-depleted follicles, despite their elevated integrin levels (*Lewellyn et al., 2013*) (*Figure 1F*). The lack of a BM stiffness gradient in non-elongating follicles is consistent with an important role for this mechanical property in organ elongation.

The data described above suggest the hypothesis that BM stiffness is in fact the anisotropic mechanical property that drives organ shape, deforming the growing tissue. An alternative hypothesis is that BM stiffness is instead an indirect consequence of organ shape, passively reflecting undetected changes in cell-intrinsic properties. To distinguish between these possibilities, we directly manipulated BM components. We then measured effects on BM mechanics and subsequent tissue elongation, including manipulations in which the A–P stiffness gradient was either eliminated

or preserved. The follicle epithelium produces most of its own BM, which can be altered by follicle-wide RNAi or by overexpression driven by *tj-Gal4* (*Figure 2I*) (*Haigo and Bilder, 2011*; *Isabella and Horne-Badovinac, 2015*; *Van De Bor et al., 2015*). AFM measurements on follicles depleted for SPARC, a factor involved in early BM incorporation of Collagen IV, showed that BM stiffness was ~80% of WT levels in the central regions, but a gradient with increased elasticity was preserved at both terminal regions and poles; elongation of these follicles was indistinguishable from that in WT follicles (*Figure 2A,B*) (*Isabella and Horne-Badovinac, 2015*; *Martinek et al., 2008*; *Pastor-Pareja and Xu, 2011*). These follicles are distinct from those that are uniformly depleted of Collagen IV, which are homogenously soft and defective in elongation, resembling *fat2-* and *msn*-depleted follicles (*Figure 2C–E*) (*Haigo and Bilder, 2011*; *Isabella and Horne-Badovinac, 2015*). By contrast, uniform overexpression of EHBP1, which elevates Collagen IV fibril deposition, leads to ~15% increased central stiffness with a ~20% increased anisotropic gradient, and results in organ hyperelongation (*Figure 2F*) (*Isabella and Horne-Badovinac, 2016*).

We then turned to spatially restricted GAL4 drivers that allow manipulation of BM components in subsets of the gradient. We depleted Collagen IV specifically in the central FCs (using *mirr-GAL4*, *Figure 2J*), where BM stiffness is normally maximal. AFM measurements showed that this manipulation eliminated stiffness differences between the central and terminal regions, and these follicles show significant elongation defects (*Figure 2G*). To complement this manipulation, we overexpressed EHBP1 locally in the terminal regions (using *fru-GAL4*, *Figure 2K*). This also equilibrated stiffness between the central and terminal regions, and again led to rounder follicles (*Figure 2H*).

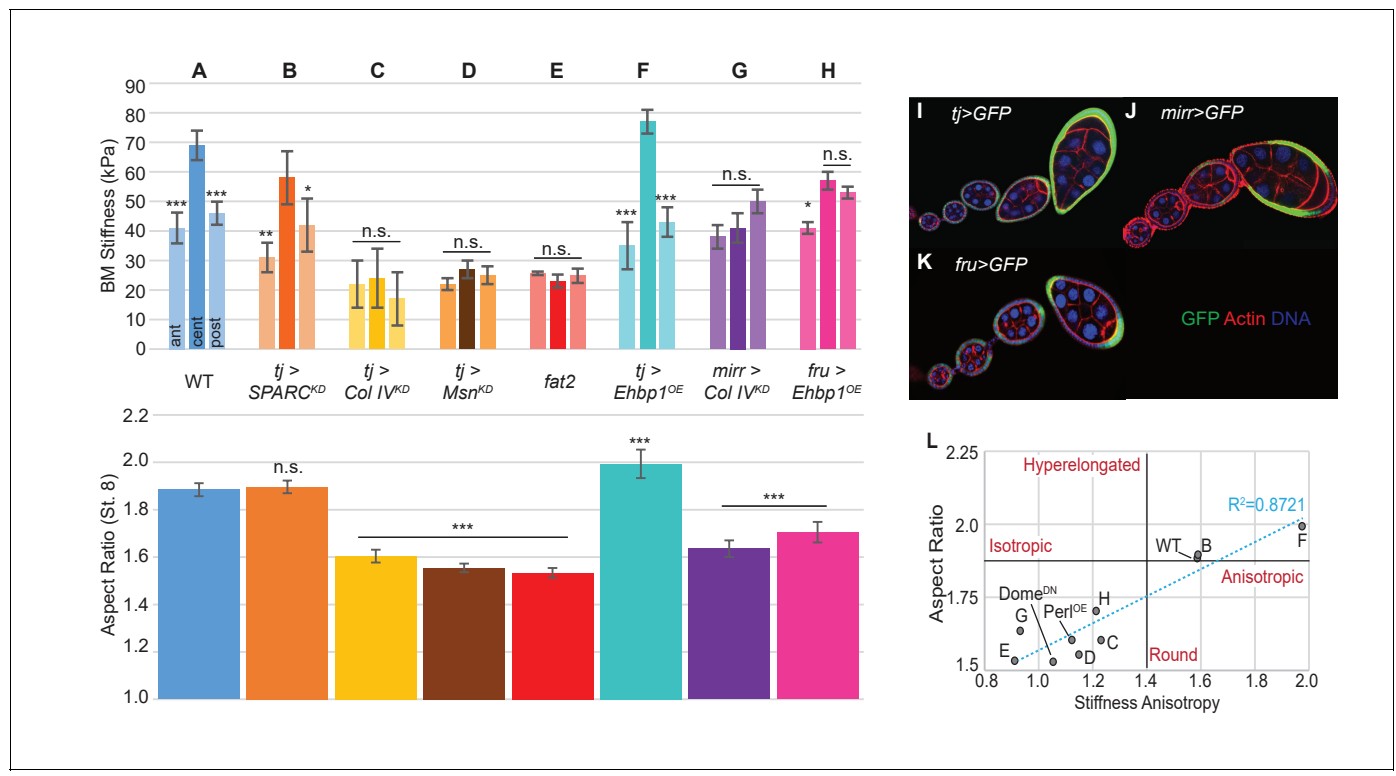

**Figure 2.** Manipulating the BM stiffness gradient alters organ shape. For each follicle genotype, AFM-measured positional stiffness at stages 7–8 is shown above and degree of elongation is shown below. Manipulations in (A–F) alter gene expression uniformly via *tjGAL4* (I) or homozygous genotype, whereas those in (G, H) alter gene expression regionally using centrally expressed *mirGAL4* or terminally expressed *fruGAL4* (J, K). Compared to WT (A), depletion of SPARC (B) softens the BM but preserves the anisotropic gradient; follicles elongate comparably to WT. Depletion of Collagen IV (ColIV) throughout the epithelium (C) creates a uniformly soft follicle with severe elongation defects, resembling mutants in which *msn* is depleted (D) or *fat2* mutants (E). EHBP1 overexpression (F) increases stiffness while retaining an anisotropic gradient, and follicles hyperelongate. Depletion of Col IV in the central region alone (G) flattens the gradient while leaving terminal stiffness intact; this results in elongation defects. EHBP1 overexpression in the terminal regions alone (H) also flattens the gradient and results in elongation defects. (L) Aspect ratio vs stiffness anisotropy (defined as the ratio of central stiffness to the mean stiffness throughout the A–P axis) for genotypes (A–H) and for *tj>Dome^DN* and *tj>Perl^OE*.

The data overall (*Figure 2L*) indicate that a spatially varying gradient in BM stiffness is essential for elongation, with absolute BM stiffness playing a lesser role. Importantly, direct manipulation of AFM-measured BM stiffness, associated with predictable changes to follicle morphogenesis, argues that the stiffness gradient is instructive for organ shape.

## Anisotropic resistance to tissue expansion by the mechanically patterned BM

To functionally test whether soft or stiff and isotropic or anisotropic BMs can indeed resist tissue expansion differentially, we adapted an organ-swelling assay (*Pastor-Pareja and Xu, 2011*). We immersed live follicles in deionized water, creating osmotic stress that leads to water influx into the follicle (*Figure 3A,B*, *Video 1*). Acute expansion of the organ challenges the BM, resulting in bursting which can be monitored by live imaging. This assay measures BM rather than epithelial failure because the follicle epithelium is disrupted well before bursting and Latrunculin A treatment does not accelerate bursting (*Figure 3C,D*). We hypothesized that the frequency and speed at which the BM bursts would reflect its overall stiffness, whereas the position at which it bursts could indicate the location of a weak point. Consistent with the former hypothesis, WT follicles at stage 8 were more resistant to bursting than those at stage 5 (*Figure 3C,D*). All collagenase-treated follicles burst instantly. Uniformly depleting Collagen IV or SPARC also induced strong increases in bursting frequency, whereas depleting Collagen IV in the central FCs alone did not (*Figure 3F*). *fat2* and *msn*-depleted follicles showed a phenotype similar to that caused by directly weakening the BM, and burst more frequently and rapidly than WT follicles (*Figure 3C,D,F*; *Videos 2* and *3*), whereas EHBP1-overexpressing follicles were completely resistant to bursting (*Figure 3F*; *Video 3*). Consistent with the latter hypothesis, WT follicles burst most frequently at polar regions, although bursting in collagenase-treated follicles showed no such preference, and *fat2* follicles burst more frequently than WT follicles in non-polar regions (*Figure 3E*). Other BM manipulations also resulted in bursting phenotypes consistent with the hypothesis (*Figure 3F,G*). For instance, depletion of Collagen IV in the central FCs (*Mirr>CoIV$^{KD}$*) relocalized swelling and bursting to this region (*Video 3*). Soft follicles generally burst more frequently and more rapidly, whereas mechanically isotropic follicles swelled more isotropically before bursting (*Figure 3F,G*). Overall, the organ-swelling experiments support the hypothesis that the WT gradient in BM stiffness provides differential resistance to organ expansion that is greatest along the central meridian, and smallest at the poles where most elongation occurs.

## Circumferential patterning of the stiffness 'corset'

In what elements does the stiffness gradient lie, and how is it generated? Previous work has suggested that the follicle is shaped by a 'molecular corset', resulting from the PCP organization of cytoskeletal elements or BM fibril-like structures (*Bilder and Haigo, 2012*; *Cetera and Horne-Badovinac, 2015*; *Gutzeit et al., 1993*; *Isabella and Horne-Badovinac, 2016*; *Tucker and Meats, 1976*). We used the 'tissue flattening' image analysis tool ImSAnE (*Heemskerk and Streichan, 2015*; *Chen et al., 2016*) to analyze follicle BM comprehensively, including BM around the entire A–P and circumferential axes of the organ (*Figure 4A*). In addition to PCP fibril organization, this approach revealed two unappreciated features.

First, around the circumferential axis, ImSAnE quantitation showed that WT follicles display a fairly uniform distribution of Collagen IV fibrils, suggesting a regular supracellular network. By contrast, in *fat2* mutant follicles, ImSAnE documented not only the loss of BM fibril polarity but also discontinuous and variable distribution of Collagen IV, with regions of high and low deposition (*Figure 4B–D*). These phenotypes were shared by follicles depleted for *msn*. Strikingly, in both *fat2*- and *msn*-depleted follicles, AFM measurements around the circumference at a single A-P position (*Figure 4E*) revealed a four-fold increase in the variability of stiffness when compared to the highly consistent stiffness of WT follicles (*Figure 4F*). The data raise the possibility that uniform circumferential mechanical properties, dependent on tissue rotation, may also be required for elongation.

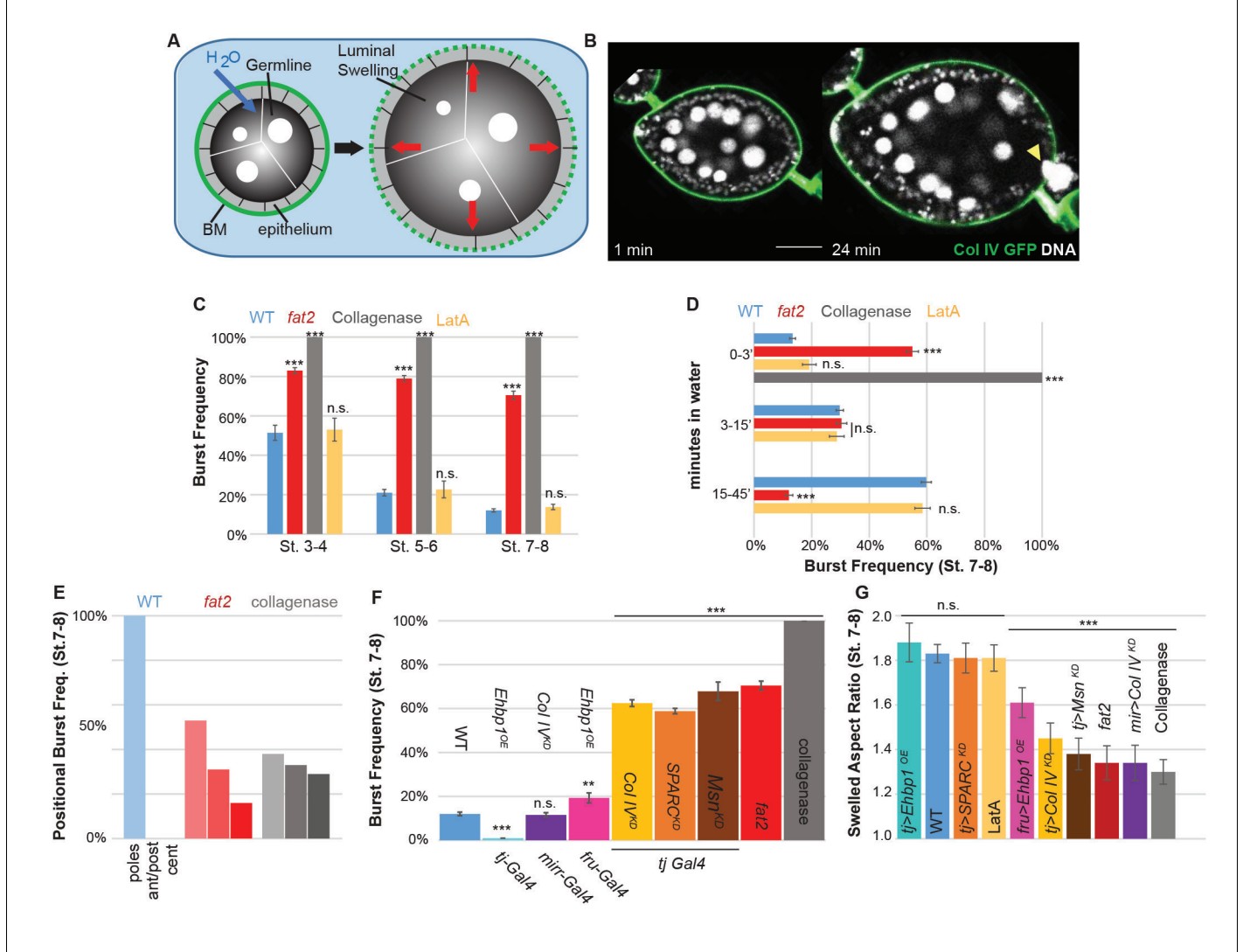

**Figure 3.** The BM stiffness gradient creates anisotropic resistance to organ expansion. (**A**) Design of osmotic-swelling experiments. Immersion in water causes influx (blue arrow) into the follicle (diagrammed in cross-section), resulting in increased turgor pressure (red arrows) that is resisted by the BM (green) as the organ swells. (**B**) WT follicle expressing ColIV–GFP, 1 min and 24 min after immersion (cf. *Video 1*). Position of the BM breach is indicated by the yellow arrowhead. (**C**) Frequency of follicle BM failure by stage and genotype, along with timing (**D**) of failure. WT BMs accommodate expansion with increasing efficiency as development proceeds in a manner independent of cellular F-actin; *fat2* and collagenased follicles burst frequently and rapidly. (**E**) Position of BM failure: WT BMs breach most frequently at the poles, whereas *fat2* and collagenased follicles also breach in other regions. (**F**) Frequency of BM failure in manipulated stage 7–8 follicles and (**G**) aspect ratio immediately before bursting. Scale bar: 25 μm.

## Morphogen-like signaling induces the organ-shaping A–P mechanical gradient

Second, along the A–P axis, we noted intriguing A–P differences in BM component levels. During elongation, Collagen IV levels are increased in central regions and taper toward the poles (*Figure 5A*). Perlecan levels, by contrast, are lower at anterior and central regions than elsewhere (*Figure 5B*). Finally, Laminin levels are fairly uniform but are low at the anterior (*Figure 5C*). We extended the analysis of Collagen IV, which is a major contributor to BM stiffness (*Morrissey and Sherwood, 2015*). Quantitation using ImSAnE documented a significant increase of Collagen IV levels in central regions as compared to anterior and posterior terminal regions (*Figure 5H,I*). This pattern is not solely transcriptional as Collagen IV subunit gene expression is not elevated in the central region (*Van De Bor et al., 2015*) (*Figure 5F*), and uniform ectopic expression of Collagen IV

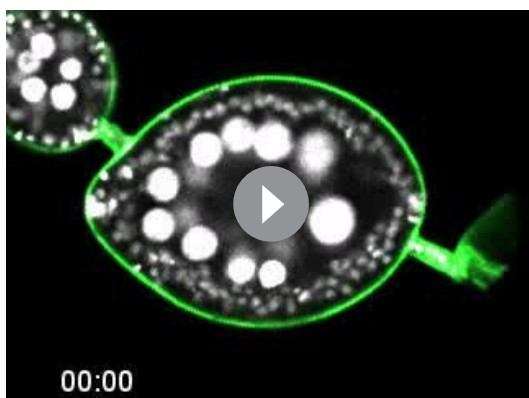

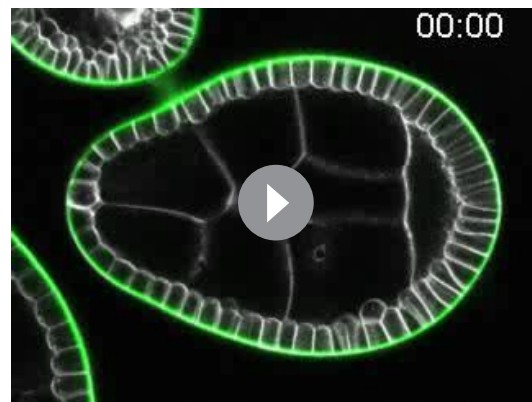

**Video 1.** WT Follicle swelling in $H_2O$. Bursting of WT follicles when placed in water as shown in *Figure 4B*. Follicle nuclei are visualized using histone–mRFP, and BM is labeled with ColIV–GFP fluorescence (green).

**Video 2.** *fat2* follicle swelling in $H_2O$. Rapid bursting of *fat2* follicles when placed in water as quantified in *Figure 4D*. The follicle is visualized using FM4-64, and BM is labeled with ColIV–GFP fluorescence (green).

subunits (via 'FLPout GAL4') results in non-uniform incorporation of Collagen IV into the BM, with enhanced levels in the follicle center (*Figure 5G*).

We investigated how these A–P differences in BM composition are regulated. Regional variance in BM stiffness will result from a combination of transcriptional and post-transcriptional regulation (including secretion, incorporation, and higher-order modification) of Collagen IV along with other BM components. We asked whether any of these processes are controlled by an organizer-like activity that exists at the follicle poles, in which secretion of a cytokine signal activates JAK/STAT to distinguish cell fates along the A–P axis (*Xi et al., 2003*). Interestingly, inhibition of JAK/STAT signaling (via expression of a dominant negative receptor) eliminated the differential A–P distribution of Collagen IV without affecting fibril polarity, and this manipulation gave rise to round follicles and eggs (*Figures 4B* and *5J*). Importantly, AFM measurements demonstrated that these follicles showed relatively high but isotropic BM stiffness (*Figure 5K*). We conclude that morphogen-like signaling results in BM mechanical patterning that drives elongation.

How do the various mechanical properties described above integrate to shape the organ? 'Molecular corset' models derive in part from analysis of follicles mutant for *fat2*, the prototypical egg elongation regulator, and their mispolarization of PCP elements such as BM fibrils (*Figure 4B*). However, *fat2* mutant follicles also fail to achieve an even distribution of BM around the follicle circumference (*Figure 4C,D*). Additionally, ImSAnE quantitation reveals that they have perturbed A–P Collagen IV pattern, although no changes in A–P signaling are seen (*Figure 5H,I*, *Figure 5—figure supplement 2*). Finally, *fat2* mutant follicles fail to undergo a whole-tissue rotation event associated with elongation (*Haigo and Bilder, 2011*; *Viktorinová and Dahmann, 2013*). To assess the role of active rotation, we depleted the actin

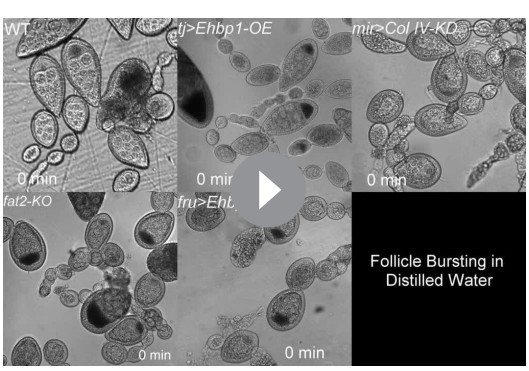

**Video 3.** Osmotic bursting of manipulated genotypes placed in water. As quantified in *Figure 4E–G*, compared to WT, *fat2* follicles burst rapidly and often not at the poles, whereas follicles uniformly overexpressing EHBP1 (*tj>EHBP1*) swell anisotropically and do not burst at all. Overexpressing EHBP1 in poles (*fru>EHBP1*) induces generally isotropic swelling but also prevents bursting. Depleting Coll IV in the central region (*mirr>Col IV KD*) cause isotropic swelling and central bursting.

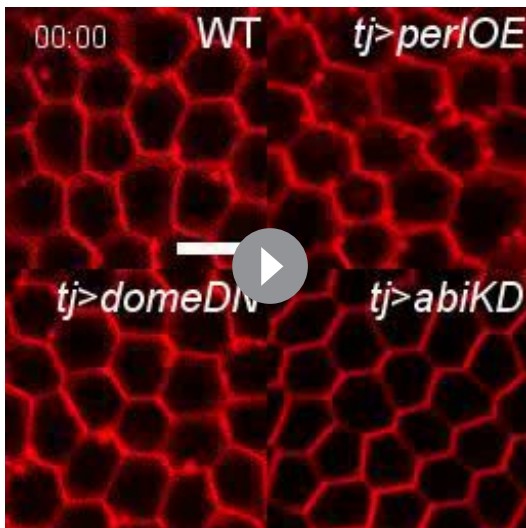

**Video 4.** Follicle rotation in manipulated genotypes. Rotation of *tj>Perl* and *tj>Dome-DN* is comparable to that of WT, whereas *tj>abi-RNAi* initiated at stage 5 blocks rotation. Scale bar: 10 μm.

regulator Abi at stage 5, which results in rotation arrest as elongation initiates (*Cetera et al., 2014*). These follicles stiffened comparably to WT, showed bursting response comparable to WT, and also elongated normally (*Figure 5- Figure 1—figure supplement 3*, *Video 4*). Conversely, elongation is prevented without disrupting rotation in several genotypes (see below, *Video 4*), confirming that phenomena other than active rotation are required to manipulate follicle shape. Nevertheless, the altered tissue-wide distributions of Collagen IV in *fat2* mutants complicate interpretations that BM fibril PCP forms the molecular corset.

We were unable to identify manipulations that independently disrupted follicle PCP and the circumferentially continuous BM distribution. Therefore, to investigate the role of BM fibril polarity per se in generating elongation-driving mechanical anisotropy, we uniformly overexpressed Perlecan, which antagonizes the constrictive properties of Collagen IV BMs and can induce round eggs (*Isabella and Horne-Badovinac, 2015*; *Pastor-Pareja and Xu, 2011*). This manipulation did not change the A–P levels, PCP, or circumferential distribution of Collagen IV fibrils (*Figures 4B,D* and *5H,I*). However, AFM analysis revealed that it did create a softer BM in which the anisotropic gradient has been eliminated, and the enclosed follicles fail to elongate (*Figure 5J,K*). In osmotic stress experiments, follicles overexpressing perlecan swelled more isotropically and burst more rapidly than WT follicles (*Figure 5L*). Thus, despite the fact that neither the levels, local PCP, or supracellular organization of Collagen IV fibrils are altered in Perlecan-overexpressing follicles, the BM of these follicles had mechanical deficits similar to those of follicles completely lacking a BM. By contrast, follicles deficient for STAT-dependent A–P signaling also fail to elongate but show normal fibril polarity and organization, and are significantly more resistant to bursting (*Figures 4B* and *5J–L*). Together, these data support a requirement for a circumferentially even distribution of PCP fibrils in elongation. However, they also reveal that PCP fibrils alone are insufficient to resist tissue growth anisotropically; the organ-shaping stiffness gradient requires patterned A–P BM levels.

## Discussion

Organ elongation is a fundamental developmental process, and is generally considered to be driven by cell-intrinsic polarized mechanical forces that actively deform tissues. Here, we demonstrate that an elongating tissue can rely instead on mechanical anisotropy patterned into the BM. Our data indicate that this asymmetric resistance within the extracellular environment, rather than asymmetric force generation within the cells, plays the dominant role in molding the follicle, prescribing subsequent morphogenetic cell behaviors. These results direct increased attention to fine BM spatial organization in creating the mechanical environment that shapes each tissue, and may fill the gap between the limited repertoire of cell-based morphogenetic mechanisms and the immense diversity of organ shapes.

Stromatic ECMs and BMs surround most animal organs, but their full roles in morphogenesis remain unresolved. Long regarded as an inert scaffold, ECM is known to influence tissue biology through actively regulating ligand availability and adhesion signaling; local BM deposition and degradation also play key roles in the branching morphogenesis of several mammalian organs (*Daley and Yamada, 2013*; *Harunaga et al., 2014*; *Morrissey and Sherwood, 2015*; *Pastor-Pareja and Xu, 2011*; *Varner and Nelson, 2014*). However, analysis of the mechanical properties of vertebrate BMs in vivo is hampered by surrounding cellular stroma, whose removal necessitates non-

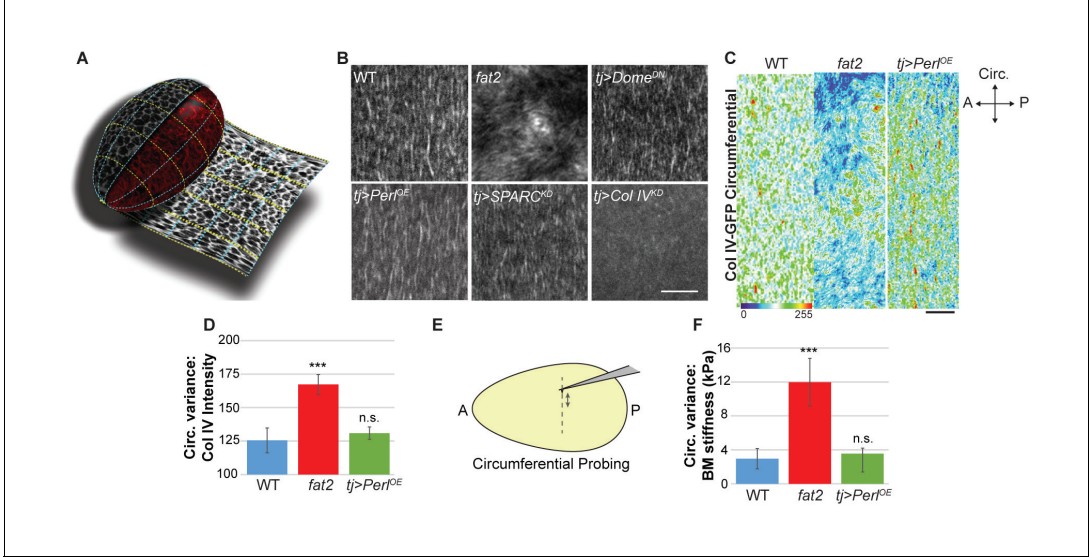

**Figure 4.** Uniform circumferential mechanics in elongating follicles. (**A**) 'Unrolling' of organ surface by ImSAnE allows quantitation of BM components along both A–P and circumferential axes. Image taken from *Chen et al. (2016*). (**B**) Analysis of BM fibril PCP shows WT polarity when Perl or Dome[DN] are overexpressed or when SPARC is depleted, contrasting with altered polarity in *fat2* and absence of polarity in Col IV-depleted mutants. (**C, D**) Unrolling reveals increased variance in circumferential Col IV levels in *fat2* as compared to those in WT or Perl-overexpressing follicles. The heat map indicates lowest (blue) to highest (red) intensities over equivalent ~35% circumferential segments. (**E, F**) AFM analysis along the circumferential axis of a follicle at a single central meridian. *fat2* mutant follicles show high variability in BM stiffness, compared to the consistent values of WT or Perl-overexpressing follicles. Scale bars: 5 µm (B) and 10 µm (C).

physiological manipulations. Because of this, only exceptionally robust BMs, such as those of the eye, have been analyzed following denuding protocols (*Ali et al., 2016*). By contrast, fly follicles lack a cellular stroma, and their topology allows direct access of AFM probes to the BM of an intact living tissue.

Our in vivo biophysical measurements of this native BM reveal an unappreciated degree of tissue-level mechanical patterning. Within each follicle, BM stiffness develops reliably and with spatial properties that are carefully regulated along both the A–P and circumferential axes. Along the A–P axis, a stiffness gradient is built that increases ~300% along a ~13-cell, 100 µm arc at stage 8. Perpendicular to this axis, stiffness around the circumference varies by less than 5% across the same distance. Our data reveal that both axes are critical for organ shaping, and merit a significant revision of the 'molecular corset' model previously proposed to mediate elongation (*Bilder and Haigo, 2012*; *Cetera and Horne-Badovinac, 2015*; *Gutzeit et al., 1993*; *Isabella and Horne-Badovinac, 2016*; *Tucker and Meats, 1976*). Hypotheses of corset structure have focused on the PCP organization of the basal actin network, the microtubule cytoskeleton, or the fibril-like BM. However, manipulations that preserve PCP alignment but nevertheless result in round follicles demonstrate that mechanical anisotropy at the length scale of individual BM fibrils is not sufficient to drive elongation. Instead, they suggest that consistent circumferential stiffness, probably associated with the supracellular BM fibrillar network generated by whole-tissue rotation, is a key element of corset effectiveness. Moreover, manipulations that flatten a pole-derived A–P signaling gradient also flatten the A–P stiffness gradient, and create isotropic organs. Thus, to drive elongation, the corset must also be anisotropic on a 'global' tissue-wide scale, in a manner that depends on morphogen-regulated mechanical properties.

The direct manipulations of BM components presented here, which lead to predicted tissue shape outcomes, argue that BM mechanics themselves are instructive for morphogenesis. Flattening the stiffness gradient in several ways, including by locally restricted BM alteration, prevents elongation, whereas hyperelongating follicles have an enhanced stiffness gradient. Although we cannot rule out undetected roles of these manipulations in altering cell behaviors via classical intercellular signaling, we see no evidence for such changes in the underlying epithelium. Instead, our results

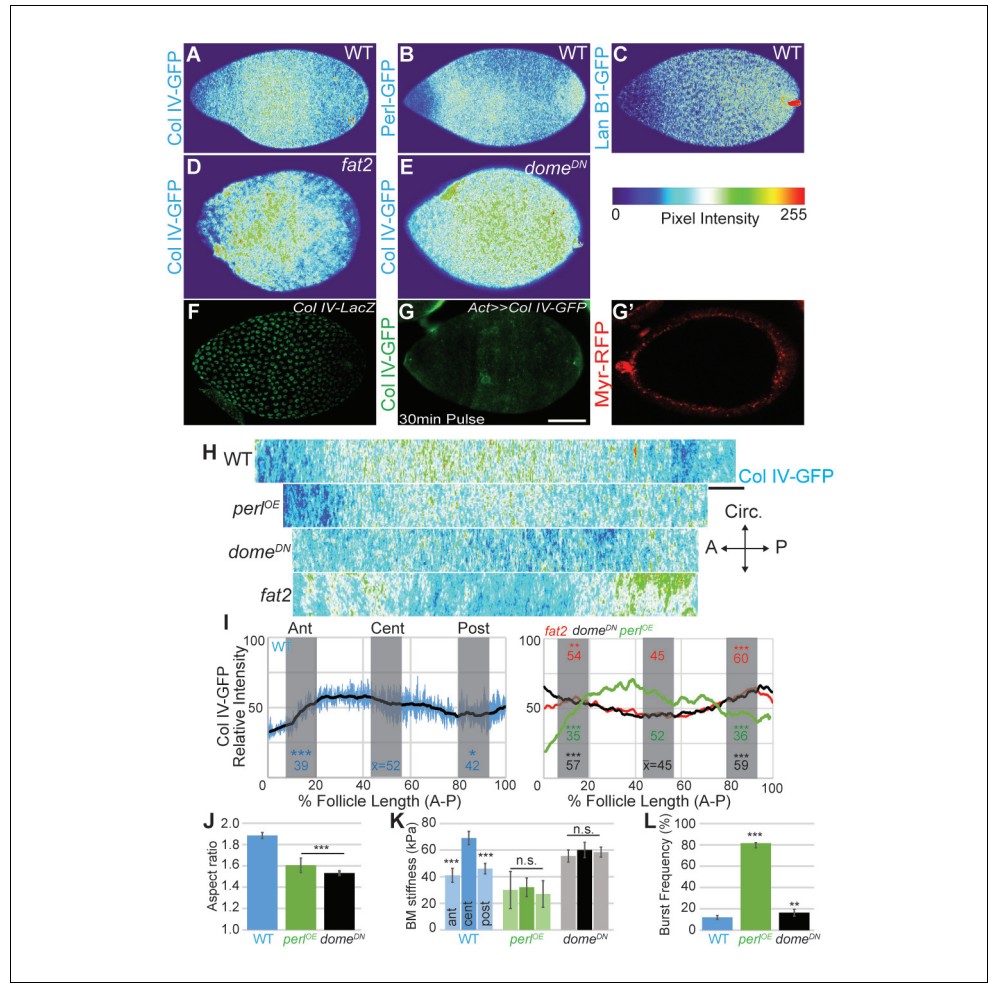

**Figure 5.** Morphogen-like signaling creates the stiffness gradient. Expression of GFP protein traps in BM components, assessed in WT stage 7–8 follicles that are physically flattened for visualization: (**A**) ColIV, (**B**) aminin B1, and (**C**) Perlecan. Heat maps indicate lowest (blue) to highest (red) intensities. The A–P ColIV pattern is disrupted in stage 7–8 follicles mutant for *fat2* ([**D**], cf. *Figure 5—figure supplement 1*) or with inhibited JAK/ STAT signaling (*tj>domeDN*, [**E**]) (cf. *Figure 5—figure supplement 2*). (**F**) Col IV transcription (*ColIV-LacZ* reporter expression) is not elevated in the central follicle. (**G**) Uniform production of ColIV (via *hsFLP; act>y+>GAL4 UAS-myr-RFP*) throughout the follicle (**G'**) results in elevated central incorporation. (**H**) ImSAnE 'unrolling' of the ColIV–GFP expressing follicle surface allows quantitation of intensity along the entire A–P axis; note the shorter axis of 'round' genotypes. (**I**) Along the A–P axis, ColIV levels are significantly elevated in the central region of WT and Perl-overexpressing follicles but not of *fat2* or *domeDN*-expressing follicles. (**J**) Elongation failure is induced by inhibition of JAK-STAT signaling or by overexpression of Perl in follicles. (**K**) AFM reveals that follicles with inhibited JAK-STAT signaling or Perl overexpression do not develop an A–P stiffness gradient; Perl overexpressing follicles are softer than WT follicles. (**L**) Perl-overexpressing follicles burst easily under osmotic challenge, whereas follicles with inhibited JAK-STAT signaling are more similar to WT. Scale bars: 25 μm (A–G') and 10 μm (H).

The following figure supplements are available for figure 5:

**Figure supplement 1.** *fat2KO* phenocopies other *fat2* null alleles.

**Figure supplement 2.** STAT reporter in *fat2* mutants.

**Figure supplement 3.** BM stiffness and active follicle rotation.

indicate that elongation is imposed by isotropic tissue growth meeting the anisotropic resistance fashioned within the BM. Consistent with this model, manipulations that alter absolute stiffness but preserve a relative gradient still result in tissue elongation. The extent to which ovarian cells respond compliantly or through well-characterized mechanical feedback mechanisms remain to be determined, but the data point to physical properties of the BM as the dominant influence.

Our results reveal a tissue elongation mechanism that is conceptually different from cell-intrinsic force asymmetries. Construction of mechanically patterned resistance in an ECM, along both axes orthogonal to its tissue interface, generates a force imbalance that imposes a specific shape on the growing organ, without necessitating spatially restricted localization of force generators within cells. Emerging examples point to the influence that substantial changes in exogenous physical forces can have in organ morphogenesis (*Aigouy et al., 2010*; *Behrndt et al., 2012*; *Etournay et al., 2015*; *Harunaga et al., 2014*; *rayRay et al., 2015*; *Shyer et al., 2013*) as well as in tumor growth (*Kaushik et al., 2016*). The discovery of precise organ-sculpting resistance within a BM motivates the development of tools and assays to explore, on a fine scale, true in vivo BM mechanical properties in both physiological and pathological contexts.

## Materials and methods

### *Drosophila* strains

The GAL4 drivers used were *tjGAL4, mirrGAL4* and *fruGAL4* (*Borensztejn et al., 2013*); *tubGAL80ts* was used to control expression temporally by shifting flies to 29°. The *Drosophila* genome contains two Collagen IV subunit-encoding genes: *CollVα1* (Flybase: *Cg25c*) and *CollVα2* (Flybase: *vkg*). For ease, both are referred to in the text and figures as Collagen IV; detailed genotypes for all experiments are listed in *Supplementary file 1*. Overexpression constructs *UAS-DT-A, UAS-Perlecan* (Flybase: *Trol*), *UAS-DomeDN*, and *UAS-EHBP1* (*Giagtzoglou et al., 2012*); RNAi constructs against *Abi, SPARC, CollVα1, CollVα2* and *msn*; and GFP protein traps in Collagen IVα2 and perlecan were obtained from the Bloomington stock center. Fosmids carrying *LanB1–GFP* (*Sarov et al., 2016*) were obtained from VDRC. *Myo–GFP* (Flybase: *sqh*) was provided by Dan Kiehart. Strains showing ectopic expression of CollV–GFP (*UAS–GFP–CollVα1 + UAS–GFP–CollVα2*) were provided by S. Noselli (*Van De Bor et al., 2015*), and utilized *hsFLP; act>y+>GAL4; UAS-myrRFP,* activated by a 30 min heat shock at 37° and immediately imaged with RFP signal to confirm uniform expression. *fat2^{KO}*, kindly provided by Mike Simon, is a null allele generated by ends-out gene replacement (*Maggert et al., 2008*) into the first exon and phenocopies other *fat2* null alleles (*Figure 5—figure supplement 1*).

### Imaging and analysis

Ovary preparations for fixed and live imaging were performed as previously described (*Chen et al., 2016*). Phalloidin-staining of fixed follicles used 20 nM phalloidin (Sigma). Latrunculin A 50 µM (Sigma), FM4-64FX 5 µg/mL (Thermo), and purified Collagenase 1000 U/mL (Worthington LS005273) were diluted in Schneider's complete media (10 mg/mL insulin, FBS and pen/strep) for live imaging. The measured osmolarity of the standard media was 300 mOsm. Hypertonic shrinking was performed in standard media supplemented with 1M D-sorbitol (Sigma) to 2000 mOsm. Fixed follicles were mounted with tape spacers, except for flattened preparations (which lacked spacers)and ImSAnE preparations (which were mounted in a depression slide). Single-plane confocal images were acquired on a Zeiss LSM700 using a Plan Apochromat 20x/NA 0.8 lens or a LD C-Apochromat 40x/NA 1.1 water-immersion lens and processed in Fiji software (*Schindelin et al., 2012*). Representative images were isolated and assembled into figures using Adobe Photoshop and Illustrator CS6.

For cortical MyoII planar polarity quantification (*Munjal et al., 2015*), IMSAnE (*Heemskerk and Streichan, 2015*) was used to 'unroll' the follicle epithelia as previously described *by Chen et al., (2016)* but with modifications. Apical surfaces of interest (SOI) of the epithelia were identified by Sqh-GFP signal. Multilayered cylinder projections of the apical-lateral membranes from the apical-most SOI plus minus 2–2.5 µm were generated by IMSAnE class CylinderMeshWrapper. Maximum intensity projections were background subtracted with the Fiji plugin 'subtract background'. A–P and circumferential junctions were categorized by 60–90° and 0–30° degrees, respectively, relative

to the A–P axis. Cortical Sqh–GFP was selected manually with line tools (width 8px) on >30 junctions of each type; the mean ratio was plotted.

For CollV–GFP intensity measurements, in toto images were collected with pixel width of 0.17 µm and voxel depth of 0.50 µm without Z-intensity correction. Follicle SOI was identified using basal F-actin signal and generalized sinusoidal projections were generated by the IMSAnE class spherelike-Fitter. Maximum intensity projections from multilayered pullbacks ±3 µm from the basal epithelia were generated. To measure A–P intensity, five 1-pixel-wide lines were drawn within a 10 µm wide stripe at the central meridian, where the pullbacks have minimal distortion. To measure circumferential intensity, five circumferential 1-pixel-wide lines were drawn within a 10 µm wide stripe along the circumferential meridian. Intensities were standardized to follicle length, then compared across follicles. Variance was calculated for each follicle using the Excel var.p formula. Profile plots were generated in Fiji software.

## Laser ablation

Ecad–GFP follicles were dissected in medium and placed in a glass-bottomed dish. A pulsed Mai-Tai two-photon on a Zeiss LSM 510 confocal microscope was used to sever A–P or circumferential junctions at anterior, central, and posterior positions on the follicle. At 708 nm and 90% power, the ablation time was less than 1 s and the resulting junction relaxation distances were measured within 300 ms. Analysis was executed manually in Fiji software normalizing the relaxed distance to the original junction length. Similar results were obtained using a UV Micropoint laser at 50% power and a Nikon Ti-E inverted microscope with a Yokogawa X1 confocal spinning disk head, with images continuously collected (500 ms/frame).

## Atomic force microscopy

BM stiffness was measured (*Figure 1—figure supplement 2*) using either a Bruker Catalyst AFM controlled by Nanoscope 8.10 software or a custom-built AFM controlled by LAbview software, both mounted on an inverted Zeiss AxioObserver Z1 microscope. MLCT-C cantilevers (Bruker) with a pyramidal tip and a nominal spring constant of 10 pN/nm were used in all experiments. The actual spring constant of each cantilever was determined by thermal calibration in air. Measurements were done in fluid. Approach velocity was optimized as 0.4 µm/sec to ensure the fastest rate of elastic measurement without viscoelastic deformation. Sample rate of deflection was 2048. Retraction speed, which does not affect elasticity measurements, was set to 20 µm/sec. Follicles were prepared as for live imaging; the cantilever was positioned at the desired position by brightfield microscopy. Each positional measurement was taken four times without moving the cantilever in XY and averaged. Young's Modulus of elasticity was calculated by fitting the cantilever deflection versus piezo extension curves to the modified Hertz model as described (*Rosenbluth et al., 2006*), using a custom-written algorithm in MATLAB (Mathworks). Only the first 50 nm of indentation were used to isolate elasticity from just the basement membrane (BM). For pole measurements, PDMS egg holders were created using custom-made molds, coated first with poly-D lysine and then treated with complete growth media. Follicles were gently mounted in PBS which was subsequently replaced with media.

## Osmotic swelling

Follicles dissected in complete media were adhered to a poly-D lysine glass-bottomed dish (MatTek) before replacing the medium twice with $dH_2O$. Images were collected at 15 s or 30 s intervals on a Zeiss Axioimager with a Plan-Neofluor 10x/0.38NA objective.

## Statistical analysis

Data were analyzed and displayed using Microsoft Excel. All error bars represent standard errors and centers represent means. At least three biological replicates were undertaken for each experiment and the results are given in *Supplementary file 1*. All acquired data were included with the exception of the AFM experiments. For these, only follicles in which all three lateral positions could be quantified were used. Statistical analysis for all data used two-tailed t-tests with p-value thresholds of *$p < 0.05$, **$p < 0.01$, and ***$p < 0.001$.

## Acknowledgements

We thank Sebastian Streichan and Jan Liphardt for helpful discussions, Jessica Feldman for generously providing access to the laser ablation microscope, Sungmin Son and Andrew Harris for help with the AFM, and Mike Simon, Stephane Noselli, Hugo Bellen, Dan Kiehart, the Vienna Drosophila Resource Center, and the Bloomington Drosophila Stock Center (NIH P40OD018537) for reagents. Laser severing experiments were conducted at the CRL Molecular Imaging Center, supported by NSF DBI-1041078. We would like to thank Holly Aaron and Jen-Yi Lee for their microscopy training and assistance. JC is a Robert Black Fellow (DRG 2173–13) and AD-M is a Fellow of the Damon Runyon Cancer Research Foundation (DRG 2157–12). This work was supported by NIH RO1s GM68675 and GM111111 to DB and GM074751 to DAF.

## Additional information

### Funding

| Funder | Grant reference number | Author |
|---|---|---|
| National Institutes of Health | GM68675 | David Bilder |
| Damon Runyon Cancer Research Foundation | DRG 2173-13 | Justin Crest |
| National Institutes of Health | GM111111 | David Bilder |
| Damon Runyon Cancer Research Foundation | DRG 2157-12 | Alba Diz-Muñoz |
| National Institutes of Health | GM074751 | Daniel A Fletcher |

The funders had no role in study design, data collection and interpretation, or the decision to submit the work for publication.

### Author contributions

JC, Conceptualization, Data curation, Formal analysis, Funding acquisition, Investigation, Methodology, Writing—original draft, Writing—review and editing; AD-M, Resources, Formal analysis, Methodology; D-YC, Software, Formal analysis, Investigation, Methodology; DAF, Conceptualization, Supervision, Funding acquisition, Writing—review and editing; DB, Conceptualization, Supervision, Funding acquisition, Methodology, Writing—original draft, Project administration, Writing—review and editing

### Author ORCIDs

Justin Crest, http://orcid.org/0000-0003-2368-1462
David Bilder, http://orcid.org/0000-0002-1842-4966

## Additional files

### Supplementary files

• Source code 1. AFM curve fitting.

• Supplementary file 1. Experimental genotypes.

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
