## [Decision Letter]

Thank you for submitting your article "Organ sculpting by patterned extracellular matrix stiffness" for consideration by *eLife*. Your article has been favorably evaluated by Marianne Bronner (Senior Editor) and three reviewers, one of whom, Allan Spradling (Reviewer #1), is a member of our Board of Reviewing Editors. The following individuals involved in review of your submission have agreed to reveal their identity: Eric F Wieschaus (Reviewer #2); Nick Brown (Reviewer #3).

Summary:

The manuscript investigates the mechanisms that restrict circumferential expansion of the *Drosophila* oocyte during its growth, thereby molding its final elongated shape. The system is specialized and builds on much previous work, but ultimately addresses a question of great generality and importance for understanding cell shape, namely whether anisotropies in (passive) resistance to an isotropic contractions or expansions are sufficient to result in polarized force distributions and morphogenetic movements. Previous works in *Drosophila* have long indicated that the follicle cells play the predominant role in shaping the oocyte. The initial expectation was that this would be a structure in the follicle cells themselves. What is exciting about the paper is the demonstration that the constraint is really in the ECM secreted by the follicles and the biophysical measurements that quantify and support that view. The demonstration that reprogrammed expression of ECM components and signaling regulators can alter stiffness is also highly significant. With several relatively minor changes described below the paper would be suitable for acceptance.

Essential revisions:

1) The initial experiments to implicate a role for the follicle cells are weak and unnecessary given previous research on this system. The remarkably crude approach of completely ablating the follicular epithelium adds nothing of value and is not currently even interpreted correctly. The ablation of follicle cell junctions along the AP and DV axis is not described in detail nor is much data provided. The failure to detect junction-mediated anisotropy does not strongly implicate the ECM because it does not exclude anisotropy in cellular structures other than junctions, such as basal actin fibers. A better approach would be to cite classical papers by Wieschaus, and by Gutzeit where mosaics are used to show that the length vs circumference ratio is governed by an ovarian cell type that is not germ line derived, most likely the follicle cells.

2) The authors should clarify whether their experiments apply to egg elongation generally or only to early elongation. The authors study the initial elongation of the follicle that takes place from about stage 3 to stage 7. However, most previous papers have studied elongation throughout oogenesis, including significant additional elongation during stages 8-14.

3) The measurement of ECM stiffness using atomic force microscopy is a great strength of the paper. However, relatively few details or controls are shown to validate these measurements. Does the size, shape and exact placement of the tip matter? A large tip might be more relevant to the large scale morphogenetic forces under consideration. The force is applied at right angles to the material, but Young's modulus is conventionally determined by the deformation along the linear dimension of the material. Do the measurements depend on the questionable assumption that the ECM is locally isotropic in all dimensions? More primary data should be shown, rather than just ratios.

4) It is not entirely clear that from the experiments presented whether additional mechanisms are needed for egg elongation beyond the reductions in collagen deposition at the poles expected for geometric reasons due to rotation. Artificially changing polar BM stiffness causes problems, but the authors’ experiments and arguments fail to delineate how precise the gradient must be, and whether graded JAK/STAT signaling is essential because of its effects on the stiffness profile or on BM synthesis. The authors should more clearly justify statements such as: "[these data]...reveal that PCP fibrils are insufficient alone to anisotropically resist tissue growth; the organ-shaping stiffness gradient requires patterned A-P BM levels".

---

## [Author Response]

*Essential revisions:*

*1) The initial experiments to implicate a role for the follicle cells are weak and unnecessary given previous research on this system. The remarkably crude approach of completely ablating the follicular epithelium adds nothing of value and is not currently even interpreted correctly. The ablation of follicle cell junctions along the AP and DV axis is not described in detail nor is much data provided. The failure to detect junction-mediated anisotropy does not strongly implicate the ECM because it does not exclude anisotropy in cellular structures other than junctions, such as basal actin fibers. A better approach would be to cite classical papers by Wieschaus, and by Gutzeit where mosaics are used to show that the length vs circumference ratio is governed by an ovarian cell type that is not germ line derived, most likely the follicle cells.*

We respect the reviewers’ criticism of these initial experiments and agree that they may detract from the more powerful experiments that follow. We have made use of published work as reviewer 2 suggested. This includes the 1981 Wieschaus paper on *short egg*, although we have previously examined *seg* mutants and cannot detect follicle shape defects before st. 8; the gene seems to regulate elongation at later stages (see point 2). Furthermore, osmotic bursting experiments revealed no significant differences between stage 8 WT and *seg* mutant follicles by (see Figure 6). We have moved former Figure 1 to a Supplemental Figure.

The cell junction ablation experiments have been updated and clarified in the Figure Legend and Experimental Methods.

Author response image 1.(**A**) St. 7-8 follicles were indented with a 10µm bead-tipped cantilever.Relative stiffness differences between A-P position were similar to pyramidal-tipped cantilevers. N=14 follicles. (**B**) St. 10 follicles (N=11) were indented and showed gradient of stiffness as in st. 7-8 (N=10), however the anterior region becomes significantly softer. (**C**) Final egg shape of selected genotypes. With the exception, of *Fru>Ehbp1^OE^*, all genotypes had defective follicle and egg elongation. N>= 20 eggs each. (**D**) Hypertonic media (2000mOsm) containing Sorbitol was used to reduce follicle osmotic pressure. Basement membrane stiffness was measured in the center of st. 7-8 follicles and was not significantly different when cultured in Sorbitol media (N= 16) compared to standard media (N=16). It is worth noting that only a very slight reduction in follicle midsagittal area was achieved with a variety of hypertonic treatments making significant conclusions problematic. (**E**) *short egg (seg,* N=63) osmotic bursting was not significantly different from WT (N=80).**DOI:**
http://dx.doi.org/10.7554/eLife.24958.020

*2) The authors should clarify whether their experiments apply to egg elongation generally or only to early elongation. The authors study the initial elongation of the follicle that takes place from about stage 3 to stage 7. However, most previous papers have studied elongation throughout oogenesis, including significant additional elongation during stages 8-14.*

The reviewers make an important point. Briefly, our goal here was to use the highly regular architecture of the early follicle to address the general question of whether basement membrane mechanics can directly instruct organ shape. Following stage 8, the non-uniform epithelial morphology, changes in nurse cell/oocyte volume, and coordinated actomyosin contractions in follicle cells create significant complications to analysis, particularly the mechanical assays at the heart of the manuscript. We note that most manipulations that cause follicle elongation defects at stage 8 also result in significant changes in final egg shape (Figure 6). We have now clarified in the manuscript that our analyses focus on the initial stage of egg elongation.

*3) The measurement of ECM stiffness using atomic force microscopy is a great strength of the paper. However, relatively few details or controls are shown to validate these measurements. Does the size, shape and exact placement of the tip matter? A large tip might be more relevant to the large scale morphogenetic forces under consideration. The force is applied at right angles to the material, but Young's modulus is conventionally determined by the deformation along the linear dimension of the material. Do the measurements depend on the questionable assumption that the ECM is locally isotropic in all dimensions? More primary data should be shown, rather than just ratios.*

Figure 2—figure supplement 2 now lays out how primary force-extension curves are used to derive stiffness scores, and the Methods have been expanded with additional detail.

We have also now repeated the measurements in WT follicles, using a cantilever with a 10 µm round bead. As shown in Figure 6, this tip also detects a mechanical gradient in the BM, with the center being significantly more stiff than the termini. Previously, the Fletcher lab examined the differences between a small pyramidal tip and a large spherical tip (Rosenbluth, Biophys J, 2006) on synthetic substrates as well as leukocytes. In brief, they found that the standard deviations of the measurements is smaller with the larger spherical beads, but the validity of the Hertzian model used to extract elasticity becomes more questionable. The pyramidal tip is therefore preferred because it can give a more accurate depiction of absolute stiffness.

Because local features of the matrix can change positional stiffness values, we made four indentations at a given position to obtain an accurate average measurement. Figure 5 shows that tip placement at multiple sites along the circumferential axis within a given position along the AP axis detects ~5% variability in stiffness. This variability is much less than the regional variability, suggesting that differences in exact tip placement do not cloud the detection of the mechanical gradient.

The AFM measurements, involving calculations of values generated by indentations normal to the plane of the material, are derived using standard approaches within the field (e.g. Radmacher, Methods in Cell Biology, 2007). Isotropy is an important assumption. As a reviewer inferred, the Hertzian model used to calculate stiffness assumes that the sample is homogenous and isotropically linearly elastic; this is nearly never the case with biological samples, and a particular concern given the apparent fibril-like structure of the follicle BM. However, to date it is the best approximation that mechanobiology has. Anisotropy of the matrix at the length scale of individual BM fibrils will certainly be interesting to investigate, but we do not currently have a good technique to do so.

*4) It is not entirely clear that from the experiments presented whether additional mechanisms are needed for egg elongation beyond the reductions in collagen deposition at the poles expected for geometric reasons due to rotation. Artificially changing polar BM stiffness causes problems, but the authors’ experiments and arguments fail to delineate how precise the gradient must be, and whether graded JAK/STAT signaling is essential because of its effects on the stiffness profile or on BM synthesis. The authors should more clearly justify statements such as: "[these data]…reveal that PCP fibrils are insufficient alone to anisotropically resist tissue growth; the organ-shaping stiffness gradient requires patterned A-P BM levels".*

Although we are struck by the relatively strong R^[2]^ between BM stiffness anisotropy and follicle aspect ration (Figure 3 and recently extended to other genotypes), unfortunately our manipulations do not yet allow us to fine-tune the stiffness gradient to answer questions about precision. As mentioned in the text, we have not yet comprehensively analyzed transcription of all BM components, but the *vkg* pattern suggests that it is not regulated by JAK/STAT signaling. However, JAK/STAT is indeed required for relatively lower incorporation of Col IV into the anterior BM, along with the lowered anterior stiffness that creates a mechanical gradient.

While we acknowledge that EHBP1 overexpression at poles could be interpreted as neomorphic stiffening, we also see follicle rounding when the physiological JAK/STAT gradient is disrupted (by Dome-DN, Figure 5). The elongation defects associated with central knockdown of Collagen IV as well as polar EHBP1 overexpression (Figure 2) suggest that a gradient is still required, even when the BM has relatively high average stiffness.

We have now revised the text to clarify our conclusions about the necessity for both the circumferential PCP fibrils and the AP incorporation gradient. The former is based on the fact that follicles overexpressing either Dome-DN or Perlecan rotate appropriately and maintain circumferential collagen organization and stiffness, but nevertheless fail to elongate. The latter of course is based on the fact that Dome-DN overexpressing follicles also flatten the AP Collagen incorporation gradient.